# The Src-Family Kinases SRC and BLK Contribute to the CLDN6-Adhesion Signaling

**DOI:** 10.3390/cells12131696

**Published:** 2023-06-23

**Authors:** Naoki Ichikawa-Tomikawa, Kotaro Sugimoto, Korehito Kashiwagi, Hideki Chiba

**Affiliations:** Department of Basic Pathology, Fukushima Medical University School of Medicine, Fukushima 960-1295, Japan; tomiknao@hirakata.kmu.ac.jp (N.I.-T.); sugikota@fmu.ac.jp (K.S.); k-kore@dokkyomed.ac.jp (K.K.)

**Keywords:** tight junction, claudin, cell adhesion signal, signal transduction, SFK, non-receptor tyrosine kinases, epithelial differentiation

## Abstract

Cell adhesion molecules, including integrins, cadherins, and claudins (CLDNs), are known to activate Src-family kinases (SFKs) that organize a variety of physiological and pathological processes; however, the underlying molecular basis remains unclear. Here, we identify the SFK members that are coupled with the CLDN6-adhesion signaling. Among SFK subtypes, BLK, FGR, HCK, and SRC were highly expressed in F9 cells and concentrated with CLDN6 along cell borders during epithelial differentiation. Immunoprecipitation assay showed that BLK and SRC, but not FGR or HCK, form a complex with CLDN6 via the C-terminal cytoplasmic domain. We also demonstrated, by pull-down assay, that recombinant BLK and SRC proteins directly bind to the C-terminal cytoplasmic domain of CLDN6 (CLDN6C). Unexpectedly, both recombinant SFK proteins recognized the CLDN6C peptide in a phosphotyrosine-independent manner. Furthermore, by comparing phenotypes of F9:*Cldn6*:*Blk*^−/−^ and F9:*Cldn6*:*Src*^−/−^ cells with those of wild-type F9 and F9:*Cldn6* cells, we revealed that BLK and SRC are essential for CLDN6-triggered cellular events, namely epithelial differentiation and the expression of retinoid acid receptor target genes. These results indicate that selective SFK members appear to participate in the CLDN-adhesion signaling.

## 1. Introduction

The members belonging to the Src-family kinases (SFKs), including BLK, FGR, FRK, FYN, HCK, LCK, LYN, SRC, and YES, are non-receptor tyrosine kinases [1,2,3,4]. They consist of the N-terminal Src-homology 4 (SH4) domain, which contains a myristoylation site for membrane localization; a unique domain; the SH3 domain, which binds proline-rich sequences; the SH2 domain, which recognizes phosphotyrosine (pY) motifs; and the kinase (SH1) domain [5,6,7,8]. SFKs are activated by the binding of respective ligands to various transmembrane-receptor-associated tyrosine kinases (RTKs), such as the epidermal growth factor receptor (EGFR), human epidermal growth factor receptor 2 (HER2), and insulin-like growth factor receptor 1 (IGF-1R). In more detail, the liganded RTKs, adaptors or effectors, are known to interact with the SH2 and/or SH3 domains of SFKs, leading to the conformational change from the inactive closed form to the active open one [7,8,9,10,11,12]. The activated SFKs phosphorylate substrates, including phosphoinositide 3-kinase (PI3K) and focal adhesion kinase (FAK), which sequentially initiate the intracellular signaling cascades and cellular responses. On the other hand, SFKs are also activated by the engagement of cell–cell or cell–matrix adhesion molecules lacking intrinsic kinase activity, such as cadherins and integrins [13,14,15,16,17]. However, information on the cell-adhesion-triggered SFK signaling is still fragmentary.

The claudin (CLDN) family members are the major transmembrane proteins of tight junctions, the apical-most apparatus of apical junctional complexes in vertebrate epithelial cell sheets [18]. They are tetra-spanning membrane proteins with a short cytoplasmic N-terminus, two extracellular loops (EC1 and EC2), and a C-terminal cytoplasmic domain. The CLDN family is composed of more than 20 members in mammals and shows distinct expression patterns in tissue- and cell-type-specific manners [19,20,21]. The fundamental function of CLDNs is to create paracellular barriers and pores for ions and substances in the corresponding EC1-dependent fashion [19,20,21,22,23,24,25,26].

CLDNs are also thought to possess signaling properties, but the underlying molecular basis has yet to be established [27]. We previously showed that the engagement of CLDN6 recruits and activates SFKs in the EC2-dependent and the C-terminal cytoplasmic-domain-dependent manners, resulting in epithelial differentiation from F9 stem cells [28,29]. We also demonstrated that both Y196 and Y200 in the first half of the C-terminal cytoplasmic domain of CLDN6 are absolutely required to activate SFKs, which in turn phosphorylate CLDN6 at Y196/200. Moreover, we found that CLDN6/SFK signal propagates the PI3K/AKT cascade, which targets the retinoic acid receptor γ (RARγ) and the estrogen receptor α (ERα) and stimulates their activities independently of the ligands. Additionally, we reported that aberrant CLDN6/SFK signaling promotes endometrial cancer progression in vitro and in vivo by hijacking the CLDN6–ERα axis [29,30,31]. Nevertheless, it remains unknown which SFK members cooperate with CLDN6 to initiate these cellular processes.

Here, we report that BLK and SRC directly bind to the C-terminal cytoplasmic domain of CLDN6 (CLDN6C). Unexpectedly, both SFK subtypes recognized the CLDN6C peptide in a pY-independent manner. We also demonstrate that BLK and SRC are indispensable for the CLDN6-adhesion signaling to induce epithelial differentiation and the expression of RAR target genes.

## 2. Materials and Methods

### 2.1. Antibodies

The antibodies used in this study are listed in Table 1.

### 2.2. Expression Vectors

The coding region of mouse *Cldn6* (full length, amino acids [AA] 1-219; Δ1/2C, AA 1-201; ΔC, AA 1-183) was amplified by RT-PCR using total RNA from F9 cells. A 3 × HA tag was added to the 5′ end of the *Cldn6* sequence using a long-tail primer. The fragments were cloned into the *Xba*I/*Kpn*I site of pD402 plasmid, in which the gene expression is driven by the *PGK* promoter. Mouse SFKs (*Blk*, *Fgr*, *Frk*, *Fyn*, *Hck*, *Lck*, *Lyn*, *Src*, and *Yes1*) were similarly amplified by PCR using cDNA of F9 cells or *Blk* Mouse Tagged ORF Clone (MR208004, OriGene, Rockville, MD, USA), and cloned into the *EcoR*I/*Sal*I site of pFLAG-CMV-5.1 (E6908, Merck, Darmstadt, Germany). To produce the recombinant proteins, *Blk*-*FLAG* and *Src*-*FLAG* were subcloned into the *KpnI/SacI* site and the *KpnI*/*HindIII* site of plEx/Bac-4 (71726, Merck), while the C-terminus of *Cldn6* (C185-V219) was inserted into the *KpnI*/*HindIII* site of plEx/Bac-4, which contains glutathione S-transferase (GST).

### 2.3. Cell Culture

F9 L32T2 cells, in which both doxycycline (Dox)-inducible gene expression and tamoxifen-dependent Cre-mediated recombination are allowed, F9:*iCldn6* (hereafter, “*i*” refers to Dox-inducible expression of a given gene), and F9:*Cldn6* cells were generated as described previously [32,33]. The cells were plated on a gelatin-coated dish and maintained in DMEM with 10% FBS. F9:*iCldn6* cells were treated with 1 µg/mL Dox (D3447, Merck) 12 h after seeding and harvested 72–96 h after treatment when epithelial differentiation was microscopically evident.

The human endometrial carcinoma cell line Ishikawa was obtained from Dr. Yamada (Wakayama Medical University), and Ishikawa:*CLDN6* cells were established as reported previously [31]. These cells were grown in DMEM with 10% FBS.

For transient expression of the target genes (*HA-Cldn6*, *HA-Cldn6Δ1/2C*, *HA-Cldn6ΔC*, *Blk-FLAG*, *Fgr-FLAG*, *Frk-FLAG*, *Fyn-FLAG*, *Hck-FLAG*, *Lck-FLAG*, *Lyn-FLAG*, *Src-FLAG*, and *Yes1-FLAG*), 1.0 × 10^6^ cells were transfected with 1 µg of the plasmids and pHRL-Puro [32] by Lipofectamine 3000 (15292465, Thermo Fisher Scientific, Waltham, MA, USA) in a 6-well plate. Twenty-four hours after transfection, the cells were treated with 0.1 µg/mL of puromycin. After 2–3 days, when the cells were grown to 70–80% confluency, the whole cell extracts were collected for immunoprecipitation. 

For gene expression analysis, the cells were maintained in charcoal-treated FBS and treated for 12 h with 1 nM of all-*trans* retinoic acid (RA; R2625, Merck) 24 h after seeding. 

### 2.4. Genome Editing

sgRNA/Cas9 all-in-one expression clones targeting mouse *Blk* and *Src* genes were obtained from GeneCopoeia (Rockville, MD, USA; MCP226646-CG01-1, MCP226646-CG01-1). The plasmids were transfected by electroporation. The cells were treated with 0.1 µg/mL of puromycin and cloned by limiting dilution. Knockout of *Blk* and *Src* genes was verified by genomic PCR and Sanger sequencing, as well as by Western blotting. The knockout clones were then transfected with pD402-*Cldn6* by electroporation as previously described [33].

### 2.5. Immunoprecipitation and Immunoblot

Total cell extracts were collected by using CellLytic MT Cell Lysis Reagent (C3228, Merck) and sonicated with three or four bursts of 5–10 s. Immunoprecipitation was performed using Immunoprecipitation Kit Protein G (11719386001, Merck) according to the manufacturer’s protocol. One microgram of ChromPure Mouse IgG (015-000-003, Jackson ImmunoResearch, West Grove, PA, USA) was used as a negative control. Whole-cell lysates or immunoprecipitated samples were mixed with sample loading buffer containing 2-mercaptoethanol and incubated for 10 min at 95 °C. They were resolved by one-dimensional SDS-PAGE and transferred onto a polyvinylidene difluoride membrane. The membranes were saturated with PVDF Blocking Reagent for Can Get Signal (NYPBR01, TOYOBO, Osaka, Japan) for 30 min. After rinsing in TBS containing 0.1% Tween 20, they were incubated with a primary antibody diluted in PBS or Can Get Signal Solution 1 (NKB-101, TOYOBO) for 1 h at room temperature or overnight at 4 °C, followed by 1 h incubation with horseradish peroxidase (HRP)-conjugated secondary antibodies diluted in PBS or Can Get Signal Solution 2 (NKB-101, TOYOBO). They were rinsed again and exposed to EzWestLumi One (WSE-7110, ATTO, Tokyo, Japan). After rinsing with 10% H_2_O_2_ to inactivate HRP, each membrane was hybridized with HRP-conjugated anti-beta actin antibody as loading controls. Whole-cell extracts from a breast cancer cell line MDA-MB-231 and 293T cells transiently overexpressing SFKs or CLDN6 were used as positive controls.

### 2.6. Immunofluorescence and Imaging

Cells were grown on a glass-based dish (3910-035, IWAKI, Shizuoka, Japan) coated with Cellmatrix Type I-A (Nitta gelatin, Osaka, Japan). The samples were fixed in 4% paraformaldehyde and 0.2% Triton-X for 10 min at room temperature. After washing with PBS, they were preincubated in PBS containing 5% skimmed milk. They were subsequently incubated overnight at 4 °C with primary antibodies diluted in Signal Booster Immunostain F (BCL-ISF, Beacle, Kyoto, Japan) and rinsed with PBS, followed by a reaction for 1 h at room temperature with appropriate secondary antibodies. All samples were examined using a laser-scanning confocal microscope (FV1000, Olympus, Tokyo, Japan). Photographs were processed with Photoshop CC (Adobe, San Jose, CA, USA) and ImageJ software (Wayne Rasband National Institutes of Health, Bethesda, MD, USA).

### 2.7. Pull-Down Assay

plEx/Bac-4, plEx/Bac-4-*Blk-FLAG*, and plEx/Bac-4-*Src-FLAG* were transfected into Sf9 insect cells using Insect GeneJuice reagent (71259, Merck). Recombinant BLK (recBLK) and recSRC proteins were obtained from these cell lysates by GST (glutathione *S*-transferase)-tagged protein purification kit (635619, Clontech, Kusatsu, Japan) according to the manufacturer’s instructions. For the GST-pull-down assay, 10 µg of the recGST, recBLK-FLAG, and recSRC proteins was bound to Glutathione-Superflow Resin (635607, Clontech) and incubated with 3 µg of recCLDN6C protein for 1 h at room temperature. The resins were washed five times in wash buffer (140 mM NaCl, 10 mM Na_2_HPO_4_, 1.8 mM KH_2_PO_4_, pH 7.5). The proteins were then eluted with SDS-PAGE sample buffer and subjected to immunoblotting under reducing conditions.

### 2.8. ELISA

A 96-well ELISA plate was coated with 500 ng of CLDN6C peptide synthesized by Eurofins Scientific, dissolved in 100 µL of TBS, and incubated overnight at 4 °C. After washing with TBS-T and blocking with 2% bovine serum albumin (BSA) at 37 °C for 30 min, recGST, recBLK (ab60876, abcam, Cambridge, UK), or recSRC (ab51424, abcam) was dissolved in TBS-T and incubated at 37 °C for 1 h. The wells were then washed three times with TBS-T and probed with horseradish peroxidase (HRP)-conjugated anti-GST antibody (sc-138 HRP, Santa Cruz Biotechnology, Dallas, TX, USA) for 1 h at room temperature. After five more washes with TBS-T, the bound HRP was detected using 3,3′,5,5′-tetramethylbenzidine (TMB) substrate (421101, BioLegend, San Diego, CA, USA). The relative absorbance to the mean value of CLDN6C/GST was plotted as a bar graph.

### 2.9. RT-qPCR

For gene expression analysis, total RNA was isolated from cells using TRIzol RNA Isolation Reagents (15596026, Thermo Fisher Scientific), and reverse transcription was performed using SentiFAST cDNA Synthesis Kit (BIO-65063, meridian BIOSCIENCE, Cincinnati, OH, USA). Quantitative PCR (qPCR) was performed using THUNDERBIRD SYBR qPCR Mix (TOYOBO) and Step One Real-Time PCR System (Applied Biosystems, Waltham, MA, USA) using the primers listed in the previous paper [28]. The expression levels of the target genes were normalized to the corresponding *Gapdh* expression.

## 3. Results

### 3.1. BLK, FGR, HCK, and SRC Are Colocalized with CLDN6 on Epithelial Cell Borders of F9 Cells

We first determined the expression profile of SFK proteins in F9:*iCldn6* cells in which CLDN6 expression is induced by Dox treatment [33]. Western blot analysis revealed that four members of SFKs (BLK, FGR HCK, and SRC) were strongly expressed in undifferentiated F9:*iCldn6* cells (Figure 1A). Although SFKs were activated in Dox-treated F9:*iCldn6* cells as expected and described previously [28], the expression levels of these SFK members were similar to those in undifferentiated F9:*iCldn6* cells. On the other hand, FRK, FYN, LCK, LYN, and YES were hardly detected in undifferentiated and differentiated F9:*iCldn6* cells. Note that we used 293T cells transfected with the corresponding mouse SFK and *Cldn6* genes as positive controls, supporting the validity of our Western blot analysis. 

Double immunofluorescence staining showed that pSFK was concentrated with CLDN6 along cell boundaries of Dox-exposed F9:*iCldn6* cells (Figure 1B), as reported previously [28]. Furthermore, BLK, FGR HCK, and SRC were colocalized with CLDN6 at mature cell–cell junctions of Dox-treated F9:*iCldn6* cells, though their positive signals were also detected in the cytoplasm and nucleus (Figure 1B). By contrast, these SFK members were scarcely concentrated along cell borders in undifferentiated F9:*iCldn6* cells (Figure 1C). It should be noted that epithelial differentiation is not induced in the entire cells because F9 cells are stem cells and therefore divide both symmetrically and asymmetrically. Colocalization of BLK and SRC with CLDN6 was also observed along cell borders in Ishikawa:*CLDN6* but not in Ishikawa cells (Appendix A). 

### 3.2. BLK and SRC Are Associated with CLDN6 through the C-Terminal Cytoplasmic Domain

We subsequently performed an immunoprecipitation assay using F9 L32T2 cells that transiently transfected with either the *Blk-FLAG*, *Fgr-FLAG*, *Hck-FLAG*, or *Src-FLAG* expression vector along with the *HA-Cldn6* expression vector. Among these SFK members, BLK and SRC, but not FGR or HCK, formed a complex with CLDN6 (Figure 2A). We previously showed that pSFKs are associated with CLDN6C [28]. Therefore, the *HA-Cldn6*, *HA-Cldn6ΔC1/2*, or *HA-Cldn6ΔC* expression vectors, in the latter two of which the second half of or the whole C-terminal cytoplasmic domain of CLDN6 is deleted, were introduced with the *Blk-FLAG* or *Src-FLAG* expression vector to F9 L32T2 cells and subjected to immunoprecipitation assay. As shown in Figure 2B, an association between CLDN6 and either BLK or SRC was evident even in the absence of the second half of the C-terminal cytoplasmic domain of CLDN6 but disappeared without the whole C-terminal cytoplasmic domain.

### 3.3. BLK and SRC Directly Bind to the C-Terminal Cytoplasmic Domain of CLDN6

We next verified, by GST pull-down assay, whether full-length recombinant BLK and SRC proteins (recGST-BLK and recGST-SRC) were able to bind directly to recCLDN6C (Figure 3A). As shown in Figure 3B, both recGST-BLK and recGST-SRC appeared to interact with recCLDN6C. Since CLDN6 recruits and activates SFKs in a Y196/200-dependent manner, as previously reported [28], we subsequently determined whether recBLK and recSRC interact with the non-pY196/200- or pY196/200-containing CLDN6C peptides that correspond to the first half of the C-terminal cytoplasmic domain of CLDN6 (Figure 4A). 

Unexpectedly, however, both recGST-BLK and recGST-SRC dose-dependently bound not only to the pY196/200-CLDN6C peptide but also to the non-pY196/200-CLDN6C peptide (Figure 4B). In addition, the binding efficiency of the pY196/200-CLDN6C peptide was significantly lower than that of the non-pY196/200-CLDN6C peptide.

### 3.4. BLK and SRC Are Essential for the CLDN6-Adhesion Signaling

To validate whether BLK and SRC are indispensable for the CLDN6-adhesion signaling, we then generated F9:*Cldn6*:*Blk*^−/−^ and F9:*Cldn6*:*Src*^−/−^ cells using CRISPR/Cas9-based genome editing and compared their phenotypes with those of wild-type (WT) F9 and F9:*Cldn6* cells. The knockout of *Blk* and *Src* genes was verified by DNA sequencing, and the absence of the corresponding protein was confirmed by Western blot (Figure 5A,B). The absence of either BLK or SRC led to a decrease in pSFK levels. The CLDN6-initiated epithelial differentiation was hindered in F9:*Cldn6*:*Blk*^−/−^ and F9:*Cldn6*:*Src*^−/−^ cells compared with F9:*Cldn6* cells (Figure 5C). In addition, the induced expression of epithelial markers (ZO-1+ variant and ezrin/radixin/moesin-binding phosphoprotein [EBP50]) was inhibited in F9:*Cldn6*:*Blk*^−/−^ and F9:*Cldn6*:*Src*^−/−^ cells (Figure 5D). Moreover, the CLDN6-triggered expression of RAR target genes Cyp26a and Cldn6 (endogenous) was completely reversed in F9:*Cldn6:Blk*^−/−^ and F9:*Cldn6:Src*^−/−^ cells (Figure 5E). Taken together, these results indicate that BLK and SRC are indispensable for mediating the CLDN6-adhesion signaling.

## 4. Discussion

We previously reported that pSFK is colocalized with CLDN6 along cell borders in F9:*Cldn6* cells, and both signals disappeared upon the SFK inhibitor PP2 treatment [28]. Colocalization of pSFK and CLDN6 was also evident at the cell–cell junctions of epithelia in embryoid bodies [28]. Moreover, we characterized the human endometrial carcinoma cell line Ishikawa expressing CLDN6, further confirming that pSFK is concentrated to cell boundaries together with CLDN6 [31]. In the present study, we demonstrated that at least four members of SFKs (BLK, FGR HCK, and SRC) are highly expressed in undifferentiated F9:*iCldn6* cells and Dox-treated F9:*iCldn6* cells. On the other hand, our double immunofluorescence staining showed that these SFK subtypes were colocalized with CLDN6 along cell borders in differentiated F9:*iCldn6* cells but not in undifferentiated F9:*iCldn6* cells. These results indicated that CLDN6 adhesion activates BLK, FGR HCK, and SRC, leading to the alteration of their subcellular localization.

We also identified, via immunoprecipitation assay, the SFK members associated with CLDN6 and obtained evidence showing that BLK and SRC form a complex with CLDN6. By contrast, neither the CLDN6-FGR nor CLDN6-HCK complexes were detected, suggesting that FGR and HCK are concentrated on cell boundaries in a CLDN6-independent manner. We subsequently demonstrated that BLK and SRC are associated with CLDN6 via the first half of the C-terminal cytoplasmic domain, in good agreement with our previous results on the CLDN6-pSFK complex in F9 cells and Ishikawa cells expressing WT and mutant CLDN6 [28]. Furthermore, GST pull-down assay revealed that BLK and SRC directly bind to the C-terminal cytoplasmic domain of CLDN6.

Another finding of the present study was that BLK and SRC bind to both non-pY196/200-CLDN6C and pY196/200-CLDN6C peptides. The binding efficiency of the pY196/200-CLDN6C peptide was lower than that of the non-pY196/200-CLDN6C peptide. These results were unexpected because most of the SH2 domains are known to recognize the pY-containing peptide better than the corresponding non-phosphorylated ones [34,35,36]. However, there are several exceptions. For instance, SH2 domains of Cten (C-terminal tensin-like), Grb7 (growth factor receptor bound 7), SAP (SLAM-associated protein), Shc (Src homologous and collagen), and Tensin2 exhibit pY-independent binding to their ligands [37,38,39,40,41,42,43,44,45,46]. Similar to these SH2 proteins, BLK and SRC seem to recognize the Y196/200-CLDN6C peptide independently of tyrosine phosphorylation. In addition, human and mouse CLDN6Y196 and CLDN6Y200 are conserved in CLDN2/3/4/7/8/10/12/14/16 and CLDN1/2/4/5/9/12/17/18, respectively [29]; therefore, it should be determined whether the C-terminal cytoplasmic domain of these CLDN subtypes could recruit specific SFK members. 

Using several SFK inhibitors (i.e., PP1, PP2, SU6656, and aminogenistein), we previously revealed that SFKs are involved in CLDN6-triggered epithelial differentiation in F9 cells and embryoid bodies [28]. In the present study, we compared phenotypes of F9:*Cldn6*:*Blk*^−/−^ and F9:*Cldn6*:*Src*^−/−^ cells with those of WT F9 and F9:Cldn6 cells and showed that BLK and SRC are essential for CLDN6-initiated cellular events such as epithelial differentiation and the expression of *Cyp26a* and *Cldn6* (endogenous) genes. Since RARγ activates *Cldn6* gene expression [28,47], the positive loop of the CLDN6/BLK–RARγ and CLDN6/SRC–RARγ pathways most likely participates not only in provoking but also in the maintenance of CLDN6-initiated cellular processes.

SFKs, including SRC, are frequently activated in many types of cancers and promote cancer progression [4,8,48,49]. We previously reported that abnormal CLDN6/SFK signaling accelerates malignant phenotypes of endometrial cancer cells via hijacking the CLDN6–ERα axis [29,31]. Excessive CLDN6/SFK signaling stimulated cell proliferation and migration, as well as tumor growth and invasion into fibrous capsules in xenografts. Furthermore, we have very recently demonstrated that aberrant CLDN4/SFK signaling drives breast cancer metabolism and progression via Liver X receptor β [50]. Hence, the appropriate and extreme CLDN/SFK signaling likely contributes to physiological and pathological processes, respectively.

## 5. Conclusions

Here, we identified that BLK and SRC are directly associated with CLDN6 through the C-terminal cytoplasmic domain. We also demonstrated that these SFKs bind to CLDN6C in a pY-independent manner. Moreover, we showed that BLK and SRC are critical for the CLDN6 signaling to trigger epithelial differentiation and the expression of RAR target genes. 

## Figures and Tables

**Figure 1 cells-12-01696-f001:**
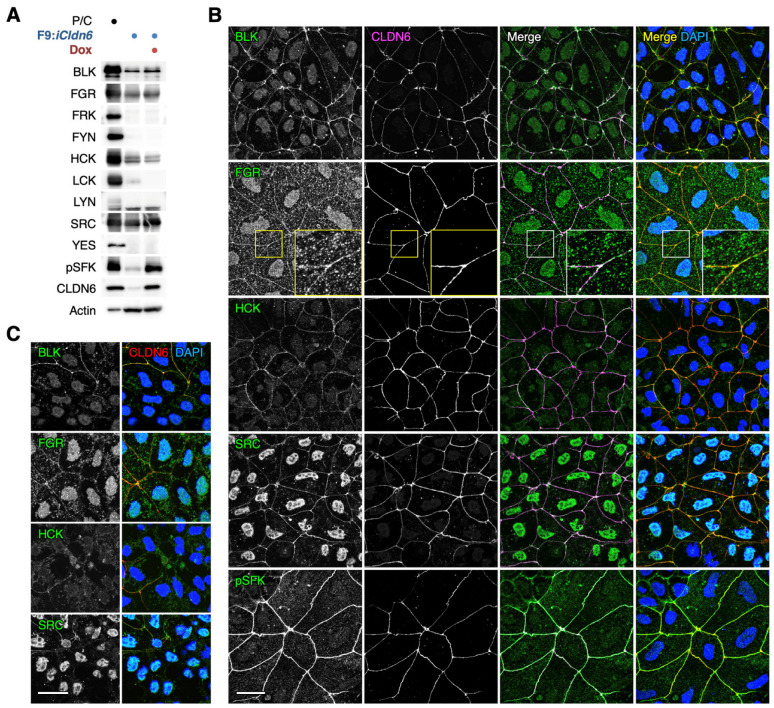
BLK, FGR, HCK, and SRC are expressed in F9 cells. (**A**) Western blot analysis showing the expression profile of SFKs in F9:*iCldn6* cells grown for 96 h in the presence or absence of 1 µg/mL doxycycline (Dox). 293T cells over-expressing the corresponding mouse *SFK* and *Cldn6* genes were used as positive controls (P/C). A breast cancer cell line MDA-MB-231 was used as the positive control for pSFK. (**B**,**C**) Confocal immunofluorescent images of the indicated proteins in Dox-exposed F9:*iCldin6* cells. Areas consisting of epithelial cells (**B**) and the boundary regions between undifferentiated and epithelial cells (**C**) are indicated. Bars, 20 µm.

**Figure 2 cells-12-01696-f002:**
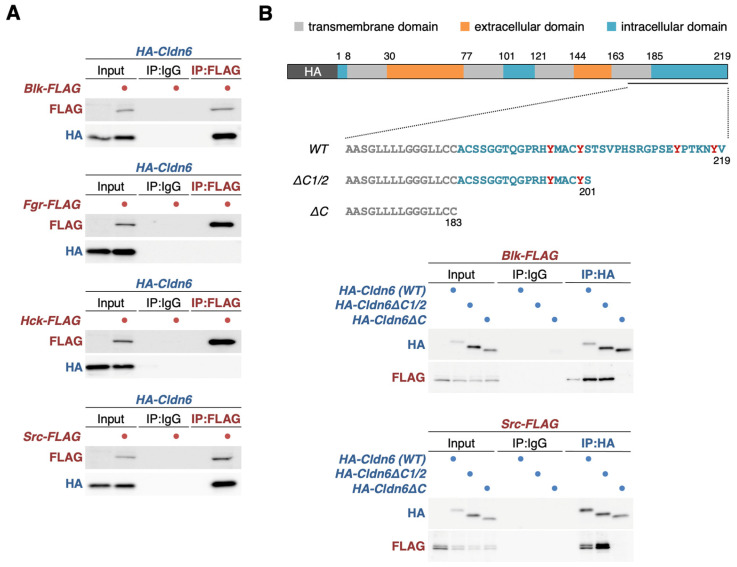
BLK and SRC form a complex with CLDN6 via the C-terminal cytoplasmic domain. (**A**) Association between CLDN6 and the indicated SFK members. (**B**) Association between CLDN6C and either BLK or SRC. F9 L32T2 cells were transiently transfected with the indicated *HA-Cldn6* expression vector along with the revealed *SFK-FLAG* expression vectors. In the input lanes, 1% of HA of the input protein samples was loaded. The schematic representation of *HA-Cldn6 (WT)* and the deletion mutants is revealed in the top panel. The conserved tyrosine residues of the C-terminal cytoplasmic domain of human CLDN6 are indicated in red. IP, immunoprecipitation; WT, wild-type.

**Figure 3 cells-12-01696-f003:**
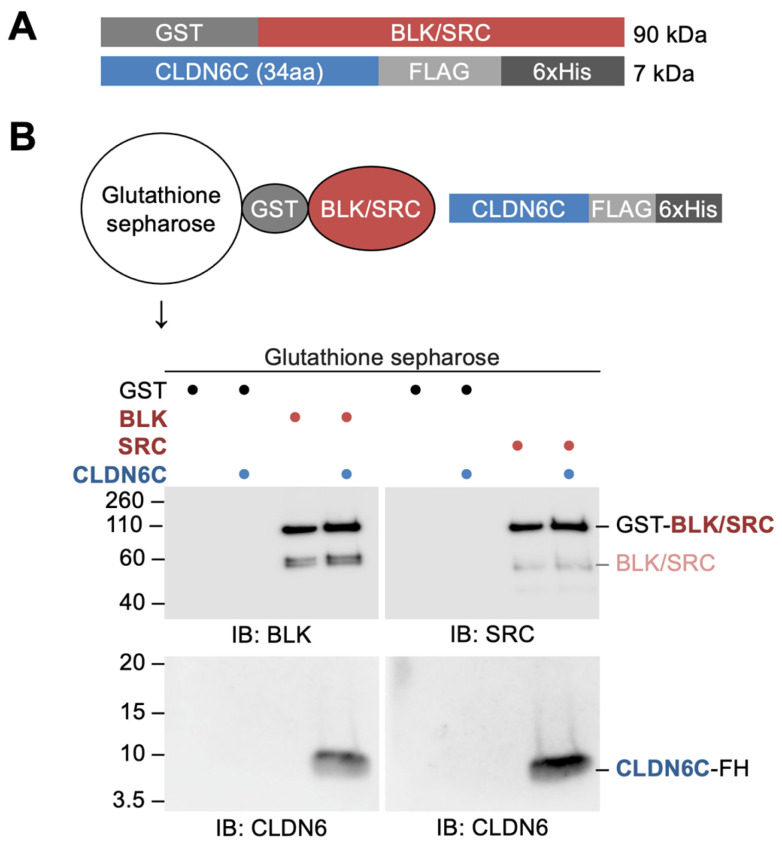
BLK and SRC directly bind to the C-terminal cytoplasmic domain of CLDN6. (**A**) The structure of recombinant BLK, SRC, and CLDN6C proteins. (**B**) GST pull-down assay showing direct binding of recombinant BLK and SRC to recombinant CLDN6C. The mobility of molecular mass markers (kilodaltons) is indicated on the left. GST, glutathione *S*-transferase; FL, FLAG; IB, immunoblot.

**Figure 4 cells-12-01696-f004:**
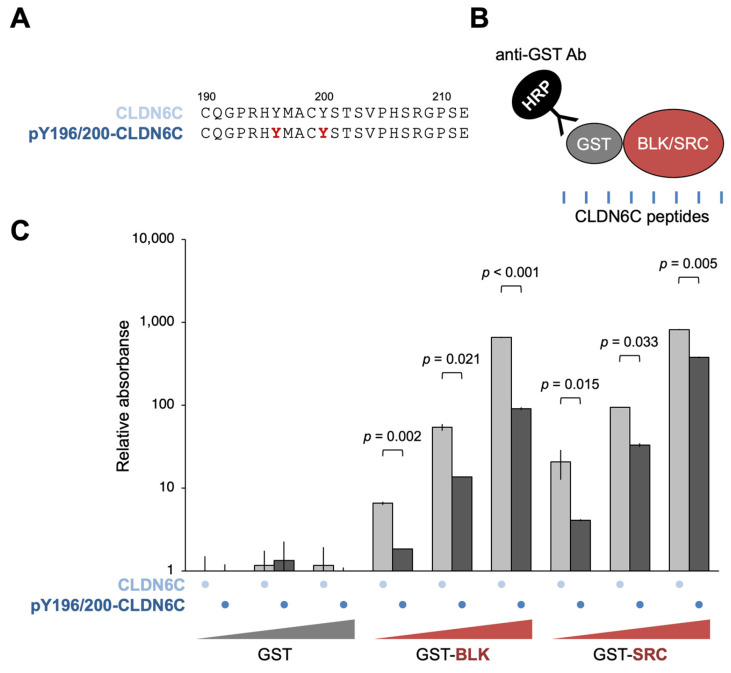
Recombinant BLK and SRC bind to the CLDN6C peptide in a pY-independent manner. (**A**) Amino acid sequences of the non-pY196/200-CLDN6C and pY196/200-CLDN6C peptides. The phosphorylation of Y residues is indicated in red. (**B**) Schema for ELISA assay. Ninety-six-well ELISA plates were coated with CLDN6C peptides and used for subsequent analysis. (**C**) ELISA analysis showing that recombinant BLK and SRC dose-dependently bind not only to the pY196/200-CLDN6C peptide but also to the non-pY196/200-CLDN6C peptide. Relative absorbance is shown in the histograms (mean ± SD; *n* = 4).

**Figure 5 cells-12-01696-f005:**
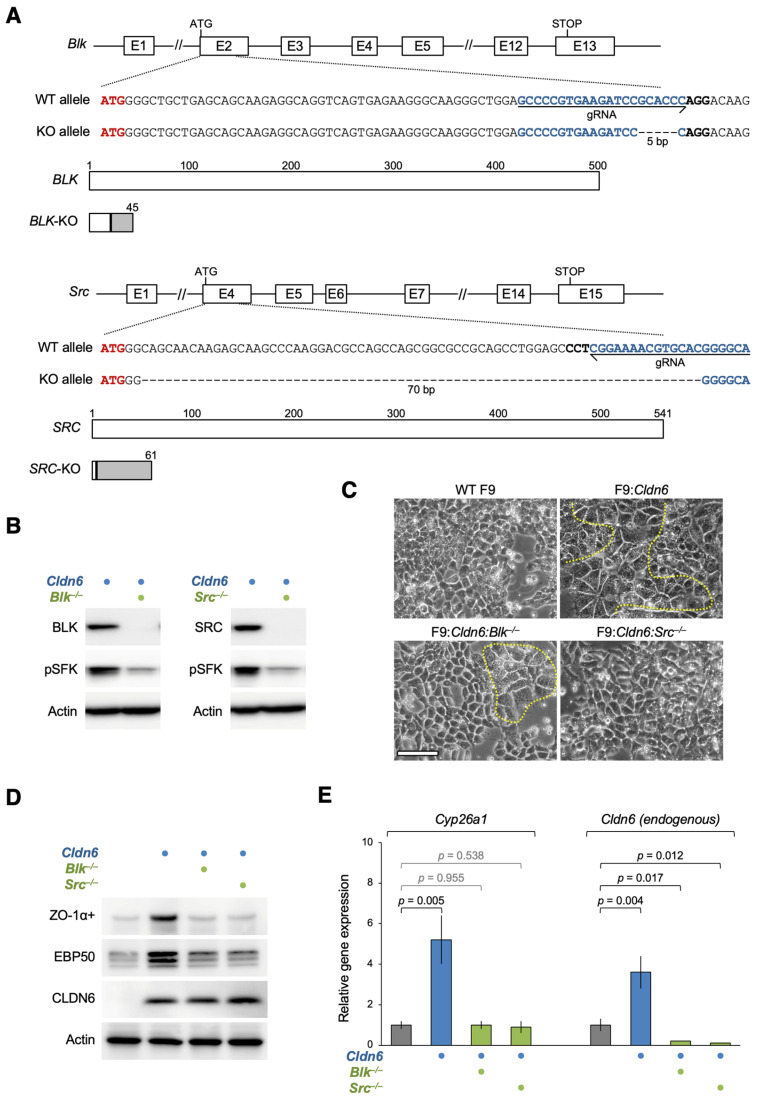
BLK and SRC are indispensable for the CLDN6-initiated cellular events. (**A**) Knockout (KO) of the *Blk* and *Src* genes in F9 cells using the CRISPR/Cas9 method. KO of these genes is verified by DNA sequencing. (**B**,**D**) Western blot for the indicated proteins in the revealed F9 cells. (**C**) Morphological appearance of the indicated F9 cells. The boundaries between undifferentiated and epithelial cells are shown by the dashed yellow lines. (**E**) RT-qPCR analysis for the indicated molecules in the revealed F9 cells. Relative expression levels are shown in the histograms (mean ± SD; *n* = 4). Bar, 50 µm.

**Table 1 cells-12-01696-t001:** Antibodies.

Antigen	Type	Host	IF	WB	IP	Source	Identifier
BLK	P	Rabbit	1/100	1/1000		Cell Signaling Technology	#3262
FGR	P	Rabbit	1/100	1/1000		Santa Cruz Biotechnology	sc-50338
FRK	M	Rat		1/1000		R&D Systems	MAB3766
FYN	P	Rabbit		1/1000		GeneTex	GTX109428
HCK	P	Rabbit	1/100	1/1000		Santa Cruz Biotechnology	sc-72
LCK	M	Mouse		1/1000		Santa Cruz Biotechnology	sc-433
LYN	P	Rabbit		1/1000		Santa Cruz Biotechnology	sc-15
SRC	M	Rabbit	1/100	1/1000		Cell Signaling Technology	#2123
YES1	M	Mouse		1/1000		Santa Cruz Biotechnology	sc-8403
pSFK	P	Rabbit	1/100	1/1000		Cell Signaling Technology	#2101
CLDN6	P	Rabbit		1/1000		Immuno-Biological Laboratories	#18865
CLDN6	M	Rat	1/4			Kojima et al. [30]	
EBP50	P	Rabbit		1/1000		Affinity Bio Reagents	PA-1-090
ZO-1 (α+ variant)	M	Rat		1/1000		Santa Cruz Biotechnology	sc33725
FLAG	M	Mouse			1 µg	MBL	M185-3L
FLAG (HRP)	P	Rabbit		1/8000		Merck	SAB4301135
HA	M	Mouse			1 µg	Proteintech	66006-2-Ig
HA (HRP)	M	Rat		1/4000		Roche	12013819001
βActin (HRP)	M	Mouse		1/4000		Santa Cruz Biotechnology	sc-47778 HRP
Mouse IgG (HRP)	P	Sheep		1/5000		GE Health Care	NA934V
Rabbit IgG (HRP)	P	Goat		1/2000		Cell Signaling Technology	#7074
Rat IgG (HRP)	P	Goat		1/2000		GE Health Care	NA935V
Rabbit IgG (Alexa 488)	P	Donkey	1/400			Jackson ImmunoResearch	711-165-152
Rabbit IgG (Alexa647)	P	Donkey	1/400			Jackson ImmunoResearch	712-605-153
Rat IgG (Cy3)	P	Donkey	1/400			Jackson ImmunoResearch	712-165-153

IF, immunofluorescence; IHC, immunohistochemistry; WB, Western blotting; IP, immunoprecipitation; M, monoclonal; P, polyclonal.

## Data Availability

The data presented in this study are available on request from the corresponding author.

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
