# Peer review of "The Src-Family Kinases SRC and BLK Contribute to the CLDN6-Adhesion Signaling"

_cells, 2023, doi:10.3390/cells12131696_

Round 1

Reviewer 1 Report

The authors showed that  BLK and SRC likely form a complex with CLDN6 via the C-terminal cytoplasmic domain in the F9:Cldn6 cells that overexpressed CLDN6.  BLK and SRC proteins likely bind to the C-terminal cytoplasmic domain of CLDN6, but independent manner with tyrosine-phosphorylation of the C-terminal domain of CLDN6. Moreover, they showed that BLK and SRC are essential for the CLDN6-triggered cellular events, such as epithelial differentiation and expression of RAR-target genes. Thus, authors concluded that SFK members, BLK and SRC, are likely important for the CLDN6-mediated cellular functions such as regulation of cell adhesion. The experiment was properly performed and  the results are nicely presented and discussed. This reviewer feel that if the authors showed more in vivo data such as mouse experiment it would further strengthen their conclusion in this study or in the future study.  The "Fig. 4E" should be corrected by "FIg. 5E" in page 8 , line 263.

Author Response

Responses to the comments of Reviewer #1

We are grateful to the reviewer’s comment “The experiment was properly performed, and the results are nicely presented and discussed”.

Specific comments

  1. This reviewer feel that if the authors showed more in vivo data such as mouse experiment it would further strengthen their conclusion in this study or in the future study.

We appreciate the constructive suggestion. As reviewer pointed, we would perform in vivo experiments in the future study. It takes approximately three months for such animal experiments to be approved by the Animal Committee at our university. Instead, we carried out an additional in vitro experiment (double immunofluorescent staining of SFK members and CLDN6 in another cell line expressing CLDN6; See Supplementary figure S1) to strengthen our conclusion.

  1. The "Fig. 4E" should be corrected by "FIg. 5E" in page 8 , line 263.

We corrected “Figure 4E” to “Figure 5E” (page 9, line 269).

Reviewer 2 Report

In this work Ichikawa-Tomikawa et al. have studied the molecular basis of claudin signalling in epithelial cells. In previous studies they have reported that claudin 6 (CLDN6) activates Src kinases. On this occasion, the authors characterize the Src kinases that could be involved in CLDN6-mediated cell differentiation. Through a combination of confocal fluorescence microscopy and biochemical analyses they identified Blk and Src as the most relevant Src kinases involved in CLDN6 signalling. They have found that both Blk and Src bind to the C-terminal domain of CLDN6 and that this binding is independent of phosphorylation of CLDN6 on Tyr196 and 200. Moreover, through CRISPR/Cas9 technology, they show that Blk and Src are important for CLDN6-mediated epithelial differentiation and RARg expression.

 The manuscript by Ichikawa-Tomikawa sheds new and interesting light on the mechanisms by which claudins are involved in the organization of epithelial layers. The manuscript is well written, experimental approaches are well done and results well performed. Only some comments and questions are arising.

 1.- The authors show that FGR and HCK localized at the cell-cell junctions in an CLDN6-independent manner. May Blk and Src also localize at intercellular junctions independently of CLDN6? To what extent is their localization at this compartment dependent on CLDN6? Fluorescence microscopy experiments of Blk and Src localization in differentiated CLD6-deficient cells are required.

2.- The authors’ finding that the interaction of Blk and Src with CLDN6 is independent of phosphorylation on Tyr is unexpected. To corroborate this result, the authors should perform this set of experiments in cells, may be using broad and/or specific Tyr phosphatase inhibitors or activators to confirm that in cells Blk/Src binding to CLDN6 is independent of phosphorylation on Tyr.

3.- Figure 2B: A schematic representation of the HA-Cldn6 deletion mutants would be helpful to envisage the domain of CLDN6 that binds Src kinases.

 4.- To what extent are Blk and Src specific partners of CLDN6? Do they bind to other claudins? Moreover, the authors mention that CLDN6 shares Tyr196 and 200 with other claudins, suggesting that they may also recruit specific SFK members. The authors should explore this issue by performing co-immunoprecipitation and protein-peptide binding assays with CLDN6 point mutants on Tyr196 and 200.

Author Response

Responses to the comments of Reviewer #2

We thank to the reviewer’s comments “The manuscript by Ichikawa-Tomikawa sheds new and interesting light on the mechanisms by which claudins are involved in the organization of epithelial layers. The manuscript is well written, experimental approaches are well done and results well performed”. We also greatly appreciate the constructive critiques that have helped to improve our manuscript.

Specific comments

  1. The authors show that FGR and HCK localized at the cell-cell junctions in an CLDN6-independent manner. May Blk and Src also localize at intercellular junctions independently of CLDN6? To what extent is their localization at this compartment dependent on CLDN6? Fluorescence microscopy experiments of Blk and Src localization in differentiated CLD6-deficient cells are required. 

To support that BLK and SRC localize at the cell-cell junctions in the CLDN6-dependent manner, we performed an additional experiment. Our double immunofluorescent study showed that BLK and SRC colocalize with CLDN6 along cell borders in Ishikawa:CLDN6 cells but not in parental Ishikawa cells (page 5, lines 200–201; Supplementary figure S1), strengthening our conclusion.

  1. The authors’ finding that the interaction of Blk and Src with CLDN6 is independent of phosphorylation on Tyr is unexpected. To corroborate this result, the authors should perform this set of experiments in cells, may be using broad and/or specific Tyr phosphatase inhibitors or activators to confirm that in cells Blk/Src binding to CLDN6 is independent of phosphorylation on Tyr.

Tyr dephosphorylation in the C-terminal end and the subsequent Tyr phosphorylation in the kinase domain of SFKs are essential for the conformational change from the inactive closed form to the active open one (References 7–12). Broad Tyr phosphatase inhibitors and activators affect not only on CLDN6 but also on SFKs in cells. Additionally, there are no CLDN6-specific Tyr phosphatase inhibitors and activators. Therefore, unfortunately, experiments using these inhibitors and activators do not corroborate our result showing that Blk/Src binding to CLDN6 is independent of Tyr phosphorylation.

  1. Figure 2B: A schematic representation of the HA-Cldn6 deletion mutants would be helpful to envisage the domain of CLDN6 that binds Src kinases.

According to the reviewer’s comment, we added a schematic representation of the WT HA-Cldn6 and the deletion mutants in Figure 2B (page 7, lines 237–238).

  1. To what extent are Blk and Src specific partners of CLDN6? Do they bind to other claudins? Moreover, the authors mention that CLDN6 shares Tyr196 and 200 with other claudins, suggesting that they may also recruit specific SFK members. The authors should explore this issue by performing co-immunoprecipitation and protein-peptide binding assays with CLDN6 point mutants on Tyr196 and 200.

We formerly performed co-immunoprecipitation experiments using Cldn6Y196A and Cldn6Y200A mutants in mouse F9 stem cells and the human endometrial carcinoma cell line Ishikawa (Sugimoto et al., PNAS, 2019; Kojima et al., 2021, Mol Cancer Res). The previous co-IP assay revealed that the CLDN6-pSFK complex is diminished in F9:Cldn6Y196A and F9:Cldn6Y200A cells, as well as in Ishikawa:Cldn6Y196A and Ishikawa:Cldn6Y200A cells, indicating that both CLDN6Y196 and CLDN6Y200 are required to recruit active SFK members (see also the attached Figures).

    On the other hand, human and mouse CLDN6Y196 and CLDN6Y200 are conserved in CLDN2/3/4/7/8/10/12/14/16 and CLDN1/2/4/5/9/12/17/18, respectively (Sugimoto and Chiba, 2021, Tissue Barriers), suggesting that these CLDN subtypes could also recruit specific SFK members. To support this possibility, we have very recently shown that CLDN4Y197, which corresponds to CLDN6Y200, are critical to activate SFKs in human breast cancer cells (Nishimagi-Murakami et al., 2023, Breast Cancer Res).

Round 2

Reviewer 2 Report

The authors have satisfactorily addressed most of my concerns and made the requested changes. The manuscript is now improved.